# The Description Length of Deep Learning Models

**Léonard Blier**
École Normale Supérieure
Paris, France
`leonard.blier@normalesup.org`

**Yann Ollivier**
Facebook Artificial Intelligence Research
Paris, France
`yol@fb.com`

## Abstract

Solomonoff's general theory of inference (Solomonoff, 1964) and the Minimum Description Length principle (Grünwald, 2007; Rissanen, 2007) formalize Occam's razor, and hold that a good model of data is a model that is good at losslessly compressing the data, including the cost of describing the model itself. Deep neural networks might seem to go against this principle given the large number of parameters to be encoded.

We demonstrate experimentally the ability of deep neural networks to compress the training data even when accounting for parameter encoding. The compression viewpoint originally motivated the use of *variational methods* in neural networks (Hinton and Van Camp, 1993; Schmidhuber, 1997). Unexpectedly, we found that these variational methods provide surprisingly poor compression bounds, despite being explicitly built to minimize such bounds. This might explain the relatively poor practical performance of variational methods in deep learning. On the other hand, simple incremental encoding methods yield excellent compression values on deep networks, vindicating Solomonoff's approach.

## 1   Introduction

Deep learning has achieved remarkable results in many different areas (LeCun et al., 2015). Still, the ability of deep models not to overfit despite their large number of parameters is not well understood. To quantify the complexity of these models in light of their generalization ability, several metrics beyond parameter-counting have been measured, such as the number of degrees of freedom of models (Gao and Jojic, 2016), or their intrinsic dimension (Li et al., 2018). These works concluded that deep learning models are significantly simpler than their numbers of parameters might suggest.

In information theory and Minimum Description Length (MDL), learning a good model of the data is recast as using the model to losslessly transmit the data in as few bits as possible. More complex models will compress the data more, but the model must be transmitted as well. The overall code-length can be understood as a combination of quality-of-fit of the model (compressed data length), together with the cost of encoding (transmitting) the model itself. For neural networks, the MDL viewpoint goes back as far as (Hinton and Van Camp, 1993), which used a variational technique to estimate the joint compressed length of data and parameters in a neural network model.

Compression is strongly related to generalization and practical performance. Standard sample complexity bounds (VC-dimension, PAC-Bayes...)  are related to the compressed length of the data in a model, and any compression scheme leads to generalization bounds (Blum and Langford, 2003). Specifically for deep learning, (Arora et al., 2018) showed that compression leads to generalization bounds (see also (Dziugaite and Roy, 2017)). Several other deep learning methods have been inspired by information theory and the compression viewpoint. In unsupervised learning, autoencoders and especially variational autoencoders (Kingma and Welling, 2013) are compression methods of the data (Ollivier, 2014). In supervised learning, the information bottleneck method studies

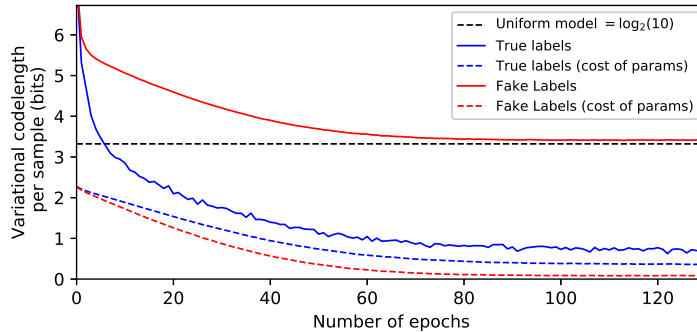

Figure 1: **Fake labels cannot be compressed** Measuring codelength while training a deep model on MNIST with true and fake labels. The model is an MLP with 3 hidden layers of size 200, with RELU units. With ordinary SGD training, the model is able to overfit random labels. The plot shows the effect of using variational learning instead, and reports the variational objective (encoding cost of the training data, see Section 3.3), on true and fake labels. We also isolated the contribution from parameter encoding in the total loss (KL term in (3.2)). With true labels, the encoding cost is below the uniform encoding, and half of the description length is information contained in the weights. With fake labels, on the contrary, the encoding cost converges to a uniform random model, with no information contained in the weights: there is no mutual information between inputs and outputs.

how the hidden representations in a neural network compress the inputs while preserving the mutual information between inputs and outputs (Tishby and Zaslavsky, 2015; Shwartz-Ziv and Tishby, 2017; Achille and Soatto, 2017).

MDL is based on Occam's razor, and on Chaitin's hypothesis that "*comprehension is compression*" (Chaitin, 2007): any regularity in the data can be exploited both to compress it and to make predictions. This is ultimately rooted in Solomonoff's general theory of inference (Solomonoff, 1964) (see also, e.g., (Hutter, 2007; Schmidhuber, 1997)), whose principle is to favor models that correspond to the "shortest program" to produce the training data, based on its Kolmogorov complexity (Li and Vitányi, 2008). If no structure is present in the data, no compression to a shorter program is possible.

The problem of overfitting fake labels is a nice illustration: convolutional neural networks commonly used for image classification are able to reach $100\%$ accuracy on random labels on the train set (Zhang et al., 2017). However, measuring the associated compression bound (Fig. 1) immediately reveals that these models do not *compress* fake labels (and indeed, theoretically, they cannot, see Appendix A), that no information is present in the model parameters, and that no learning has occurred.

In this work we explicitly measure how much current deep models actually compress data. (We introduce no new architectures or learning procedures.) As seen above, this may clarify several issues around generalization and measures of model complexity. Our contributions are:

- We show that the traditional method to estimate MDL codelengths in deep learning, variational inference (Hinton and Van Camp, 1993), yields surprisingly inefficient codelengths for deep models, despite explicitly minimizing this criterion. This might explain why variational inference as a regularization method often does not reach optimal test performance.

- We introduce new practical ways to compute tight compression bounds in deep learning models, based on the MDL toolbox (Grünwald, 2007; Rissanen, 2007). We show that *prequential coding* on top of standard learning, yields much better codelengths than variational inference, correlating better with test set performance. Thus, despite their many parameters, deep learning models do compress the data well, even when accounting for the cost of describing the model.

## 2 Probabilistic Models, Compression, and Information Theory

Imagine that Alice wants to efficiently transmit some information to Bob. Alice has a dataset $\mathcal{D} = \{(x_1, y_1), ..., (x_n, y_n)\}$ where $x_1, ..., x_n$ are some inputs and $y_1, ..., y_n$ some labels. We do not assume that these data come from a "true" probability distribution. Bob also has the data $x_1, ..., x_n$, but he does not have the labels. This describes a *supervised learning* situation in which the inputs $x$ may be publicly available, and a prediction of the labels $y$ is needed. How can deep learning models help with data encoding? One key problem is that Bob does not necessarily know the precise, trained model that Alice is using. So some explicit or implicit transmission of the model itself is required.

We study, in turn, various methods to encode the labels $y$, with or without a deep learning model. Encoding the labels knowing the inputs is equivalent to estimating their mutual information (Section 2.4); this is distinct from the problem of practical network compression (Section 3.2) or from using neural networks for lossy data compression. Our running example will be image classification on the MNIST (LeCun et al., 1998) and CIFAR10 (Krizhevsky, 2009) datasets.

### 2.1 Definitions and notation

Let $\mathcal{X}$ be the input space and $\mathcal{Y}$ the output (label) space. In this work, we only consider classification tasks, so $\mathcal{Y} = \{1, ..., K\}$. The dataset is $\mathcal{D} := \{(x_1, y_1), ..., (y_n, x_n)\}$. Denote $x_{k:l} := (x_k, x_{k+1}, ..., x_{l-1}, x_l)$. We define a *model* for the supervised learning problem as a conditional probability distribution $p(y|x)$, namely, a function such that for each $x \in \mathcal{X}$, $\sum_{y \in \mathcal{Y}} p(y|x) = 1$. A *model class*, or *architecture*, is a set of models depending on some parameter $\theta$: $\mathcal{M} = \{p_\theta, \theta \in \Theta\}$. The *Kullback–Leibler divergence* between two distributions is $\mathrm{KL}(\mu\|\nu) = \mathbb{E}_{X\sim\mu}[\log_2 \frac{\mu(x)}{\nu(x)}]$.

### 2.2 Models and codelengths

We recall a basic result of compression theory (Shannon, 1948).

**Proposition 1** (Shannon–Huffman code). *Suppose that Alice and Bob have agreed in advance on a model $p$, and both know the inputs $x_{1:n}$. Then there exists a code to transmit the labels $y_{1:n}$ losslessly with codelength (up to at most one bit on the whole sequence)*

$$L_p(y_{1:n}|x_{1:n}) = -\sum_{i=1}^{n} \log_2 p(y_i|x_i) \tag{2.1}$$

This bound is known to be optimal if the data are independent and coming from the model $p$ (Mackay, 2003). The one additional bit in the Shannon–Huffman code is incurred only once for the whole dataset (Mackay, 2003). With large datasets this is negligible. Thus, from now on we will systematically omit the $+1$ as well as admit non-integer codelengths (Grünwald, 2007). We will use the terms *codelength* or *compression bound* interchangeably.

This bound is exactly the categorical *cross-entropy loss* evaluated on the model $p$. Hence, trying to minimize the description length of the outputs over the parameters of a model class is equivalent to minimizing the usual classification loss.

Here we do not consider the practical implementation of compression algorithms: we only care about the theoretical *bit length* of their associated encodings. We are interested in measuring the amount of information contained in the data, the mutual information between input and output, and how it is captured by the model. Thus, we will directly work with codelength functions.

An obvious limitation of the bound (2.1) is that Alice and Bob both have to know the model $p$ in advance. This is problematic if the model must be learned from the data.

### 2.3 Uniform encoding

The uniform distribution $p_{\mathrm{unif}}(y|x) = \frac{1}{K}$ over the $K$ classes does not require any learning from the data, thus no additional information has to be transmitted. Using $p_{\mathrm{unif}}(y|x)$ (2.1) yields a codelength

$$L^{\mathrm{unif}}(y_{1:n}|x_{1:n}) = n \log_2 K \tag{2.2}$$

Table 1: **Compression bounds via Deep Learning.** Compression bounds given by different codes on two datasets, MNIST and CIFAR10. The *Codelength* is the number of bits necessary to send the labels to someone who already has the inputs. This codelength *includes* the description length of the model. The *compression ratio* for a given code is the ratio between its codelength and the codelength of the uniform code. The *test accuracy* of a model is the accuracy of its predictions on the test set. For 2-part and network compression codes, we report results from (Han et al., 2015a) and (Xu et al., 2017), and for the intrinsic dimension code, results from (Li et al., 2018). The values in the table for these codelengths and compression ratio are lower bounds, only taking into account the codelength of the weights, and not the codelength of the data encoded with the model (the final loss is not always available in these publications). For variational and prequential codes, we selected the model and hyperparameters providing the best compression bound.

| CODE | MNIST | | | CIFAR10 | | |
|---|---|---|---|---|---|---|
| | CODELENGTH (kbits) | COMP. RATIO | TEST ACC | CODELENGTH (kbits) | COMP. RATIO | TEST ACC |
| UNIFORM | 199 | 1. | 10% | 166 | 1. | 10% |
| FLOAT32 2-PART | $> 8.6$Mb | $> 45.$ | 98.4% | $> 428$Mb | $> 2500.$ | **92.9%** |
| NETWORK COMPR. | $> 400$ | $> 2.$ | 98.4% | $> 14$Mb | $> 83.$ | **93.3%** |
| INTRINSIC DIM. | $> 9.28$ | $> 0.05$ | 90% | $> 92, 8$ | $> 0.56$ | 70% |
| VARIATIONAL | 22.2 | 0.11 | 98.2% | 89.0 | 0.54 | 66,5% |
| PREQUENTIAL | **4.10** | **0.02** | **99.5%** | **45.3** | 0.27 | **93.3%** |

This *uniform encoding* will be a sanity check against which to compare the other encodings in this text. For MNIST, the uniform encoding cost is $60000 \times \log_2 10 = 199$ kbits. For CIFAR, the uniform encoding cost is $50000 \times \log_2 10 = 166$ kbits.

## 2.4 Mutual information between inputs and outputs

Intuitively, the only way to beat a trivial encoding of the outputs is to use the mutual information (in a loose sense) between the inputs and outputs.

This can be formalized as follows. Assume that the inputs and outputs follow a "true" joint distribution $q(x, y)$. Then any transmission method with codelength $L$ satisfies (Mackay, 2003)

$$\mathbb{E}_q[L(y|x)] \geq H(y|x) \tag{2.3}$$

Therefore, the gain (per data point) between the codelength $L$ and the trivial codelength $H(y)$ is

$$H(y) - \mathbb{E}_q[L(y|x)] \leq H(y) - H(y|x) = I(y; x) \tag{2.4}$$

the mutual information between inputs and outputs (Mackay, 2003).

Thus, the gain of *any* codelength compared to the uniform code is limited by the amount of mutual information between input and output. (This bound is reached with the true model $q(y|x)$.) Any successful compression of the labels is, at the same time, a direct estimation of the mutual information between input and output. The latter is the central quantity in the Information Bottleneck approach to deep learning models (Shwartz-Ziv and Tishby, 2017).

Note that this still makes sense without assuming a true underlying probabilistic model, by replacing the mutual information $H(y) - H(y|x)$ with the "absolute" mutual information $K(y) - K(y|x)$ based on Kolmogorov complexity $K$ (Li and Vitányi, 2008).

## 3 Compression Bounds via Deep Learning

Various compression methods from the MDL toolbox can be used on deep learning models. (Note that a given model can be stored or encoded in several ways, some of which may have large codelengths. A good model in the MDL sense is one that admits at least one good encoding.)

## 3.1 Two-Part Encodings

Alice and Bob can first agree on a model class (such as "neural networks with two layers and 1,000 neurons per layer"). However, Bob does not have access to the labels, so Bob cannot train the parameters of the model. Therefore, if Alice wants to use such a parametric model, the parameters themselves have to be transmitted. Such codings in which Alice first transmits the parameters of a model, then encodes the data using this parameter, have been called *two-part codes* (Grünwald, 2007).

**Definition 1** (Two-part codes). Assume that Alice and Bob have first agreed on a model class $(p_\theta)_{\theta \in \Theta}$. Let $L_{\text{param}}(\theta)$ be any encoding scheme for parameters $\theta \in \Theta$. Let $\theta^*$ be any parameter. The corresponding *two-part codelength* is

$$L_{\theta^*}^{\text{2-part}}(y_{1:n}|x_{1:n}) := L_{\text{param}}(\theta^*) + L_{p_{\theta^*}}(y_{1:n}|x_{1:n}) = L_{\text{param}}(\theta^*) - \sum_{i=1}^{n} \log_2 p_{\theta^*}(y_i|x_i) \quad (3.1)$$

An obvious possible code $L_{\text{param}}$ for $\theta$ is the standard float32 binary encoding for $\theta$, for which $L_{\text{param}}(\theta) = 32 \dim(\theta)$. In deep learning, two-part codes are widely inefficient and much worse than the uniform encoding (Graves, 2011). For a model with 1 million parameters, the two-part code with float32 binary encoding will amount to $32 \,\text{Mbits}$, or 200 times the uniform encoding on CIFAR10.

## 3.2 Network Compression

The practical encoding of trained models is a well-developed research topic, e.g., for use on small devices such as cell phones. Such encodings can be seen as two-part codes using a clever code for $\theta$ instead of encoding every parameter on 32 bits. Possible strategies include training a *student layer* to approximate a well-trained network (Ba and Caruana, 2014; Romero et al., 2015), or pipelines involving retraining, pruning, and quantization of the model weights (Han et al., 2015a,b; Simonyan and Zisserman, 2014; Louizos et al., 2017; See et al., 2016; Ullrich et al., 2017).

Still, the resulting codelengths (for compressing the labels given the data) are way above the uniform compression bound for image classification (Table 1).

Another scheme for network compression, less used in practice but very informative, is to sample a random low-dimensional affine subspace in parameter space and to optimize in this subspace (Li et al., 2018). The number of parameters is thus reduced to the dimension of the subspace and we can use the associated two-part encoding. (The random subspace can be transmitted via a pseudo-random seed.) Our methodology to derive compression bounds from (Li et al., 2018) is detailed in Appendix B.

## 3.3 Variational and Bayesian Codes

Another strategy for encoding weights with a limited precision is to represent these weights by random variables: the uncertainty on $\theta$ represents the precision with which $\theta$ is transmitted. The *variational code* turns this into an explicit encoding scheme, thanks to the *bits-back* argument (Honkela and Valpola, 2004). Initially a way to compute codelength bounds with neural networks (Hinton and Van Camp, 1993), this is now often seen as a regularization technique (Blundell et al., 2015). This method yields the following codelength.

**Definition 2** (Variational code). Assume that Alice and Bob have agreed on a model class $(p_\theta)_{\theta \in \Theta}$ and a prior $\alpha$ over $\Theta$. Then for any distribution $\beta$ over $\Theta$, there exists an encoding with codelength

$$L_{\beta}^{\text{var}}(y_{1:n}|x_{1:n}) = \text{KL}\left(\beta \| \alpha\right) + \mathbb{E}_{\theta \sim \beta}\left[L_{p_\theta}(y_{1:n}|x_{1:n})\right] = \text{KL}\left(\beta \| \alpha\right) - \mathbb{E}_{\theta \sim \beta}\left[\sum_{i=1}^{n} \log_2 p_\theta(y_i|x_i)\right]$$
$$(3.2)$$

This can be minimized over $\beta$, by choosing a parametric model class $(\beta_\phi)_{\phi \in \Phi}$, and minimizing (3.2) over $\phi$. A common model class for $\beta$ is the set of multivariate Gaussian distributions $\{\mathcal{N}(\mu, \Sigma), \mu \in \mathbb{R}^d, \Sigma \text{ diagonal}\}$, and $\mu$ and $\Sigma$ can be optimized with a stochastic gradient descent algorithm (Graves, 2011; Kucukelbir et al., 2017). $\Sigma$ can be interpreted as the precision with which the parameters are encoded.

The variational bound $L_\beta^{\mathrm{var}}$ is an upper bound for the Bayesian description length bound of the Bayesian model $p_\theta$ with parameter $\theta$ and prior $\alpha$. Considering the Bayesian distribution of $y$,

$$p_{\mathrm{Bayes}}(y_{1:n}|x_{1:n}) = \int_{\theta \in \Theta} p_\theta(y_{1:n}|x_{1:n})\alpha(\theta)d\theta, \tag{3.3}$$

then Proposition 1 provides an associated code via (2.1) with model $p_{\mathrm{Bayes}}$: $L^{\mathrm{Bayes}}(y_{1:n}|x_{1:n}) = -\log_2 p_{\mathrm{Bayes}}(y_{1:n}|x_{1:n})$ Then, for any $\beta$ we have (Graves, 2011)

$$L_\beta^{\mathrm{var}}(y_{1:n}|x_{1:n}) \geq L^{\mathrm{Bayes}}(y_{1:n}|x_{1:n}) \tag{3.4}$$

with equality if and only if $\beta$ is equal to the Bayesian posterior $p_{\mathrm{Bayes}}(\theta|x_{1:n}, y_{1:n})$. Variational methods can be used as approximate Bayesian inference for intractable Bayesian posteriors.

We computed practical compression bounds with variational methods on MNIST and CIFAR10. Neural networks that give the best variational compression bounds appear to be smaller than networks trained the usual way. We tested various fully connected networks and convolutional networks (Appendix C): the models that gave the best variational compression bounds were small LeNet-like networks. To test the link between compression and test accuracy, in Table 1 we report the best model based on compression, not test accuracy. This results in a drop of test accuracy with respect to other settings.

On MNIST, this provides a codelength of the labels (knowing the inputs) of $24.1\,\mathrm{kbits}$, i.e., a compression ratio of $0.12$. The corresponding model achieved $95.5\%$ accuracy on the test set.

On CIFAR, we obtained a codelength of $89.0\,\mathrm{kbits}$, i.e., a compression ratio of $0.54$. The corresponding model achieved $61.6\%$ classification accuracy on the test set.

We can make two observations. First, choosing the model class which minimizes variational codelength selects smaller deep learning models than would cross-validation. Second, the model with best variational codelength has low classification accuracy on the test set on MNIST and CIFAR, compared to models trained in a non-variational way. This aligns with a common criticism of Bayesian methods as too conservative for model selection compared with cross-validation (Rissanen et al., 1992; Foster and George, 1994; Barron and Yang, 1999; Grünwald, 2007).

### 3.4 Prequential or Online Code

The next coding procedure shows that deep neural models which generalize well also compress well.

The prequential (or online) code is a way to encode both the model and the labels without *directly* encoding the weights, based on the *prequential approach to statistics* (Dawid, 1984), by using *prediction strategies*. Intuitively, a model with default values is used to encode the first few data; then the model is trained on these few encoded data; this partially trained model is used to encode the next data; then the model is retrained on all data encoded so far; and so on.

Precisely, we call $p$ a *prediction strategy* for predicting the labels in $\mathcal{Y}$ knowing the inputs in $\mathcal{X}$ if for all $k$, $p(y_{k+1}|x_{1:k+1}, y_{1:k})$ is a conditional model; namely, any strategy for predicting the $k+1$-label after already having seen $k$ input-output pairs. In particular, such a model may *learn* from the first $k$ data samples. Any prediction strategy $p$ defines a model on the whole dataset:

$$p^{\mathrm{preq}}(y_{1:n}|x_{1:n}) = p(y_1|x_1) \cdot p(y_2|x_{1:2}, y_1) \cdot \ldots \cdot p(y_n|x_{1:n}, y_{1:n-1}) \tag{3.5}$$

Let $(p_\theta)_{\theta \in \Theta}$ be a deep learning model. We assume that we have a learning algorithm which computes, from any number of data samples $(x_{1:k}, y_{1:k})$, a trained parameter vector $\hat{\theta}(x_{1:k}, y_{1:k})$. Then the data is encoded in an incremental way: at each step $k$, $\hat{\theta}(x_{1:k}, y_{1:k})$ is used to predict $y_{k+1}$.

In practice, the learning procedure $\hat{\theta}$ may only reset and retrain the network at certain timesteps. We choose timesteps $1 = t_0 < t_1 < \ldots < t_S = n$, and we encode the data by blocks, always using the model learned from the already transmitted data (Algorithm 2 in Appendix D). A uniform encoding is used for the first few points. (Even though the encoding procedure is called "online", it does not mean that only the most recent sample is used to update the parameter $\hat{\theta}$: the optimization procedure $\hat{\theta}$ can be any predefined technique using all the previous samples $(x_{1:k}, y_{1:k})$, only requiring that the algorithm has an explicit stopping criterion.) This yields the following description length:

**Definition 3** (Prequential code). Given a model $p_\theta$, a learning algorithm $\hat{\theta}(x_{1:k}, y_{1:k})$, and retraining timesteps $1 = t_0 < t_1 < ... < t_S = n$, the *prequential* codelength is

$$L^{\text{preq}}(y_{1:n}|x_{1:n}) = t_1 \log_2 K + \sum_{s=0}^{S-1} -\log_2 p_{\hat{\theta}_{t_s}}(y_{t_s+1:t_{s+1}}|x_{t_s+1:t_{s+1}}) \qquad (3.6)$$

where for each $s$, $\hat{\theta}_{t_s} = \hat{\theta}(x_{1:t_s}, y_{1:t_s})$ is the parameter learned on data samples 1 to $t_s$.

The model parameters are never encoded explicitly in this method. The difference between the prequential codelength $L^{\text{preq}}(y_{1:n}|x_{1:n})$ and the log-loss $\sum_{t=1}^n -\log_2 p_{\hat{\theta}_{t_K}}(y_t|x_t)$ of the final trained model, can be interpreted as the amount of information that the trained parameters contain about the data contained: the former is the data codelength if Bob does not know the parameters, while the latter is the codelength of the same data knowing the parameters.

Prequential codes depend on the performance of the underlying training algorithm, and take advantage of the model's generalization ability from the previous data to the next. In particular, the model training should yield good generalization performance from data $[1; t_s]$ to data $[t_s + 1; t_{s+1}]$.

In practice, optimization procedures for neural networks may be stochastic (initial values, dropout, data augmentation...), and Alice and Bob need to make all the same random actions in order to get the same final model. A possibility is to agree on a random seed $\omega$ (or pseudorandom numbers) beforehand, so that the random optimization procedure $\hat{\theta}(x_{1:t_s}, y_{1:t_s})$ is deterministic given $\omega$, Hyperparameters may also be transmitted first (the cost of sending a few numbers is small).

Prequential coding with deep models provides excellent compression bounds. On MNIST, we computed the description length of the labels with different networks (Appendix D). The best compression bound was given by a convolutional network of depth 8. It achieved a description length of $4.10\,\text{kbits}$, i.e., a compression ratio of $0.021$, with $99.5\%$ test set accuracy (Table 1). This codelength is 6 times smaller than the variational codelength.

On CIFAR, we tested a simple multilayer perceptron, a shallow network, a small convolutional network, and a VGG convolutional network (Simonyan and Zisserman, 2014) first without data augmentation or batch normalization (VGGa) (Ioffe and Szegedy, 2015), then with both of them (VGGb) (Appendix D). The results are in Figure 2. The best compression bound was obtained with VGGb, achieving a codelength of $45.3\,\text{kbits}$, i.e., a compression ratio of $0.27$, and $93\%$ test set accuracy (Table 1). This codelength is twice smaller than the variational codelength. The difference between VGGa and VGGb also shows the impact of the training procedure on codelengths for a given architecture.

**Model Switching.** A weakness of prequential codes is the *catch-up phenomenon* (Van Erven et al., 2012). Large architectures might overfit during the first steps of the prequential encoding, when the model is trained with few data samples. Thus the encoding cost of the first packs of data might be worse than with the uniform code. Even after the encoding cost on current labels becomes lower, the cumulated codelength may need a lot of time to "catch up" on its initial lag. This can be observed in practice with neural networks: in Fig. 2, the VGGb model needs 5,000 samples on CIFAR to reach a cumulative compression ratio $< 1$, even though the encoding cost per label becomes drops below uniform after just 1,000 samples. This is efficiently solved by *switching* (Van Erven et al., 2012) between models (see Appendix E). Switching further improves the practical compression bounds, even when just switching between copies of the same model with different SGD stopping times (Fig. 3, Table 2).

## 4  Discussion

**Too Many Parameters in Deep Learning Models?**  >From an information theory perspective, the goal of a model is to extract as much mutual information between the labels and inputs as possible—equivalently (Section 2.4), to compress the labels. This cannot be achieved with 2-part codes or practical network compression. With the variational code, the models do compress the data, but with a worse prediction performance: one could conclude that deep learning models that achieve the best prediction performance cannot compress the data.

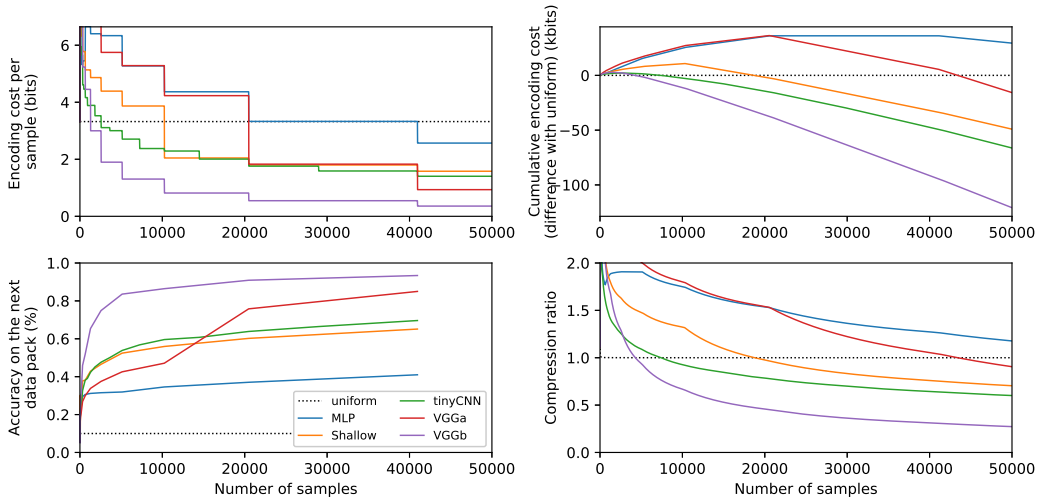

Figure 2: **Prequential code results on CIFAR.** Results of prequential encoding on CIFAR with 5 different models: a small Multilayer Perceptron (MLP), a shallow network, a small convolutional layer (tinyCNN), a VGG-like network without data augmentation and batch normalization (VGGa) and the same VGG-like architecture with data augmentation and batch normalization (VGGb) (see Appendix D). Performance is reported during online training, as a function of the number of samples seen so far. Top left: codelength per sample (log loss) on a pack of data $[t_k; t_{k+1})$ given data $[1; t_k)$. Bottom left: test accuracy on a pack of data $[t_k; t_{k+1})$ given data $[1; t_k)$, as a function of $t_k$. Top right: difference between the prequential cumulated codelength on data $[1; t_k]$, and the uniform encoding. Bottom right: compression ratio of the prequential code on data $[1; t_k]$.

Thanks to the prequential code, we have seen that deep learning models, even with a large number of parameters, compress the data well: from an information theory point of view, *the number of parameters is not an obstacle to compression*. This is consistent with Chaitin's hypothesis that "*comprehension is compression*", contrary to previous observations with the variational code.

**Prequential Code and Generalization.** The prequential encoding shows that a model that generalizes well for every dataset size, will compress well. The efficiency of the prequential code is directly due to the generalization ability of the model at each time.

Theoretically, three of the codes (two-parts, Bayesian, and prequential based on a maximum likelihood or MAP estimator) are known to be asymptotically equivalent under strong assumptions ($d$-dimensional *identifiable* model, data coming from the model, suitable Bayesian prior, and technical assumptions ensuring the effective dimension of the trained model is not lower than $d$): in that case, these three methods yield a codelength $L(y_{1:n}|x_{1:n}) = nH(Y|X) + \frac{d}{2}\log_2 n + \mathcal{O}(1)$ (Grünwald, 2007). This corresponds to the BIC criterion for model selection. Hence there was no obvious reason for the prequential code to be an order of magnitude better than the others.

However, deep learning models do not usually satisfy *any* of these hypotheses. Moreover, our prequential codes are not based on the maximum likelihood estimator at each step, but on standard deep learning methods (so training is regularized at least by dropout and early stopping).

**Inefficiency of Variational Models for Deep Networks.** The objective of variational methods is equivalent to minimizing a description length. Thus, on our image classification tasks, variational methods do not have good results *even for their own objective*, compared to prequential codes. This makes their relatively poor results at test time less surprising.

Understanding this observed inefficiency of variational methods is an open problem. As stated in (3.4), the variational codelength is an upper bound for the Bayesian codelength. More precisely,

$$L_\beta^{\mathrm{var}}(y_{1:n}|x_{1:n}) = L^{\mathrm{Bayes}}(y_{1:n}|x_{1:n}) + \mathrm{KL}\left(p_{\mathrm{Bayes}}(\theta|x_{1:n}, y_{1:n})\|\beta\right) \qquad (4.1)$$

with notation as above, and with $p_{\mathrm{Bayes}}(\theta|x_{1:n}, y_{1:n})$ the Bayesian posterior on $\theta$ given the data. Empirically, on MNIST and CIFAR, we observe that $L^{\mathrm{preq}}(y_{1:n}|x_{1:n}) \ll L_{\beta}^{\mathrm{var}}(y_{1:n}|x_{1:n})$.

Several phenomena could contribute to this gap. First, the optimization of the parameters $\phi$ of the approximate Bayesian posterior might be imperfect. Second, even the optimal distribution $\beta^*$ in the variational class might not approximate the posterior $p_{\mathrm{Bayes}}(\theta|x_{1:n}, y_{1:n})$ well, leading to a large KL term in (4.1); this would be a problem with the choice of variational posterior class $\beta$. On the other hand we do not expect the choice of Bayesian prior to be a key factor: we tested Gaussian priors with various variances as well as a conjugate Gaussian prior, with similar results. Moreover, Gaussian initializations and L2 weight decay (acting like a Gaussian prior) are common in deep learning. Finally, the (untractable) Bayesian codelength based on the exact posterior might itself be larger than the prequential codelength. This would be a problem of underfitting with parametric Bayesian inference, perhaps related to the catch-up phenomenon or to the known conservatism of Bayesian model selection (end of Section 3.3).

## 5   Conclusion

Deep learning models can represent the data *together with the model* in fewer bits than a naive encoding, despite their many parameters. However, we were surprised to observe that variational inference, though explicitly designed to minimize such codelengths, provides very poor such values compared to a simple incremental coding scheme. Understanding this limitation of variational inference is a topic for future research.

## Acknowledgments

First, we would like to thank the reviewers for their careful reading and their questions and comments. We would also like to thank Corentin Tallec for his technical help, and David Lopez-Paz, Moustapha Cissé, Gaétan Marceau Caron and Jonathan Laurent for their helpful comments and advice.

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
