[Supplementary Material · Appendix.pdf]

# A  Fake labels are not compressible

In the introduction, we stated that fake labels could not be compressed. This means that the optimal codelength for this labels is *almost* the uniform one. This can be formalized as follows. We define a *code* for $y_{1:n}$ as any program (in a reference Turing machine) that outputs $y_{1:n}$, and denote $L(y_{1:n})$ the length of this program, or $L(y_{1:n}|x_{1:n})$ for programs that may use $x_{1:n}$ as their input.

**Proposition 2.** *Assume that $x_1, ..., x_n$ are inputs, and that $Y_1, ..., Y_n$ are iid random labels uniformly sampled in $\{1, ..., K\}$. Then for any $\delta \in \mathbb{N}^*$, with probability $1 - 2^{-\delta}$ the values $Y_1, \ldots, Y_n$ satisfy that for any possible coding procedure $L$ (even depending on the values of $x_{1:n}$), the codelength of $Y_{1:n}$ is at least*

$$L(Y_{1:n}|x_{1:n}) \geq nH(Y) - \delta - 1 \tag{A.1}$$
$$= n \log_2 K - \delta - 1. \tag{A.2}$$

We insist that this does not require any assumptions on the coding procedure used, so this result holds for all possible models. Moreover, this is really a property of the sampled values $Y_1, \ldots Y_n$: most values of $Y_{1:n}$ can just not be compressed by any algorithm.

*Proof.* This proposition is a standard counting argument, or an immediate consequence of Theorem 2.2.1 in (Li and Vitányi, 2008). Let $\mathcal{A} = \{1, ..., K\}^n$ be the set of all possible outcomes for the sequence of random labels. We have $|\mathcal{A}| = K^n$. Let $\delta$ be an integer, $\delta \in \mathbb{N}^*$, we want to know how many elements in $\mathcal{A}$ can be encoded in less than $\log_2 |\mathcal{A}| - \delta$ bits. We consider, on a given Turing machine, the number of programs of length less than $\lfloor \log_2 |\mathcal{A}| - \delta \rfloor$. This number is less than :

$$\sum_{i=0}^{\lfloor \log_2 |\mathcal{A}| \rfloor - \delta - 1} 2^i = 2^{\lfloor \log_2 |\mathcal{A}| \rfloor - \delta} - 1 \tag{A.3}$$
$$\leq 2^{-\delta}|\mathcal{A}| - 1 \tag{A.4}$$

Therefore, the number of elements in $\mathcal{A}$ which can be described in less than $\log_2 |\mathcal{A}| - \delta$ bits is less than $2^{-\delta}|\mathcal{A}| - 1$. We can deduce from this that the number of elements in $\mathcal{A}$ which cannot be described by *any* program in less than $2^{-\delta}|\mathcal{A}| - 1$ bits is at least $|\mathcal{A}|(1 - 2^{-\delta})$. Equivalently, there are at least $|\mathcal{A}|(1 - 2^{-\delta})$ elements $(y_1, ..., y_n)$ in $|\mathcal{A}|$ such that for any coding scheme, $L(y_{1:n}|x_{1:n}) \geq n \log_2 K - \delta - 1$. Since the random labels $Y_1, ..., Y_n$ are uniformly distributed, the result follows. $\square$

# B  Technical details on compression bounds with random affine subspaces

We describe in Algorithm 1 the detailed procedure which allows to compute compression bounds with the random affine subspace method (Li et al., 2018). To compute the numerical results in Table 1, we took the *intrinsic dimension* computed in the original paper, and considered that the precision of the parameter was 32 bits, following the authors' suggestion. Then, the description length of the model itself is $32\times$ the intrinsic dimension. This does not take into account the description length of the labels given the model, which is non-negligible (to take this quantity into account, we would need to know the loss on the training set of the model, which was not specified in the original paper). Thus we only get a lower bound.

---

**Algorithm 1** Encoding with random affine subspaces

---

Alice transmits a parametric model $(p_\theta)_{\theta \in \Theta}$.

Alice transmits the random seed $\omega$ (if using stochastic optimization), and a dimension $k$.

Alice and Bob both sample a random affine subspace $\tilde{\Theta} \subset \Theta$, with the seed $\omega$. This means that they sample $\theta_0$ and a matrix $W$ of dimension $k \times d$ where $d$ is the dimension of $\Theta$. It defines a new parametric model $\tilde{p}_\phi = p_{\theta_0 + W \cdot \phi}$

Alice optimizes the parameter $\phi^*$ with a gradient descent algoritm in order to minimize $-\log_2 \tilde{p}_\phi(y_{1:n}|x_{1:n})$.

Alice sends $\phi^*$ with a precision $\varepsilon$ to Bob. It costs $k \times \log_2 \varepsilon$.

Alice sends the labels $y_{1:n}$ with the models $\tilde{p}_{\phi^*}$. It costs $-\log_2 \tilde{p}_{\phi^*}(y_{1:n}|x_{1:n})$

---

For MNIST, the model with the smaller intrinsic dimension is the LeNet, which has an intrinsic dimension of 290 for an accuracy of 90% (the threshold at which (Li et al., 2018) stop by definition, hence the performance in Table 1). This leads to a description length for the model of 9280 bits, which corresponds to a 0.05 compression ratio, without taking into account the description length of the labels given the model.

For CIFAR, again with the LeNet architecture, the intrinsic dimension is 2,900. This leads to a description length for the model of 92800 bits, which corresponds to a 0.05 compression ratio, without taking into account the description length of the labels given the model.

These bounds could be improved by optimizing the precision $\varepsilon$. Indeed, reducing the precision makes the model less accurate and increases the encoding cost of the labels with the model, but it decreases the encoding cost of the parameters. Therefore, we could find an optimal precision $\varepsilon^*$ to improve the compression bound. This would be a topic for future work.

## C   Technical Details on Variational Learning for Section 3.3

Variational learning was performed using the library Pyvarinf (Tallec and Blier, 2018).

We used a prior $\alpha = \mathcal{N}(0, \sigma_0^2 I_d)$ with $\sigma_0 = 0.05$, chosen to optimize the compression bounds.

The chosen class of posterior was the class of multivariate gaussian distributions with diagonal covariance matrix $\{\mathcal{N}(\mu, \Sigma) \, , \, \mu \in \mathbb{R}^d \, \Sigma \text{ diagonal}\}$. It was parametrized by $(\beta_{\mu,\rho})_{(\mu,\rho) \in \mathbb{R}^d \times \mathbb{R}^d}$, with $\sigma \in \mathbb{R}^d$ defined as $\sigma_i = \log(1 + \exp(\rho_i))$, and the covariance matrix $\Sigma$ as the diagonal matrix with diagonal values $\sigma_1^2, ..., \sigma_d^2$.

We optimize the bound (3.2) as a function of $(\mu, \rho)$ with a gradient descent method, and estimate its values and gradient with a Monte-Carlo method. Since the prior and posteriors are gaussian, we have an explicit formula for the first part of the variational loss $\mathrm{KL}(\beta_{\mu,\rho} \| \alpha)$ (Hinton and Van Camp, 1993). Therefore, we can easily compute its values and gradients. For the second part

$$(\mu, \rho) \to \mathbb{E}_{\theta \sim \beta_{\mu,\rho}}\left[ \sum_{i=1}^{n} -\log_2 p_\theta(y_i|x_i) \right], \tag{C.1}$$

we can use the following proposition (Graves, 2011). For any function $f \colon \Theta \to \mathbb{R}$, we have

$$\frac{\partial}{\partial \mu_i} \mathbb{E}_{\theta \sim \beta_{\mu,\rho}}[f(\theta)] = \mathbb{E}_{\theta \sim \beta_{\mu,\rho}}\left[ \frac{\partial f}{\partial \theta_i}(\theta) \right] \tag{C.2}$$

$$\frac{\partial}{\partial \rho_i} \mathbb{E}_{\theta \sim \beta_{\mu,\rho}}[f(\theta)] = \frac{\partial \sigma_i}{\partial \rho_i} \cdot \mathbb{E}_{\theta \sim \beta_{\mu,\rho}}\left[ \frac{\partial f}{\partial \theta_i} \cdot \frac{\theta_i - \mu_i}{\sigma_i} \right] \tag{C.3}$$

Therefore, we can estimate the values and gradients of (3.2) with a Monte-Carlo algorithm:

$$\frac{\partial}{\partial \mu_i} \mathbb{E}_{\theta \sim \beta_{\mu,\rho}}[f(\theta)] \approx \sum_{s=1}^{S} \frac{\partial f}{\partial \theta_i}(\theta^s) \tag{C.4}$$

$$\frac{\partial}{\partial \rho_i} \mathbb{E}_{\theta \sim \beta_{\mu,\rho}}[f(\theta)] \approx \frac{\partial \sigma_i}{\partial \rho_i} \cdot \sum_{s=1}^{S} \frac{\partial f}{\partial \theta_i}(\theta^s) \cdot \frac{\theta_i^s - \mu_i}{\sigma_i} \tag{C.5}$$

where $\theta^1, ..., \theta^S$ are sampled from $\beta_{\mu,\rho}$. In practice, we used $S = 1$ both for the computations of the variational loss and its gradients.

We used both convolutional and fully connected architectures, but in our experiments fully connected models were better for compression. For CIFAR and MNIST, we used fully connected networks with two hidden layers of width 256, trained with SGD, with a 0.005 learning rate and mini-batchs of size 128.

For CIFAR and MNIST, we used a LeNet-like network with 2 convolutional layers with 6 and 16 filters, both with kernels of size 5 and 3 fully connected layers. Each convolutional is followed by a ReLU activation and a max-pooling layer. The code will be publicly available. The first and the second fully connected layers are of dimension 120 and 84 and are followed by ReLU activations. The last one is followed by a softmax activation layer. The code for all models will be publicly available.

During the test phase, we sampled parameters $\hat{\theta}$ from the learned distribution $\beta$, and used the model $p_{\hat{\theta}}$ for prediction. This explains why our test accuracy on MNIST is lower than other numerical results (Blundell et al., 2015), since they use for prediction the averaged model with parameters $\hat{\theta} = \mathbb{E}_{\theta \sim \beta_{m,r}}[\theta] = \mu$. But our goal was not to get the best prediction score, but to evaluate the model which was used for compression on the test set.

# D  Technical details on prequential learning

**Prequential Learning on MNIST.**  On MNIST, we used three different models:

1. The uniform probability over the labels.
2. A fully connected network or Multilayer Perceptron (MLP) with two hidden layers of dimension 256.
3. A VGG-like convolutional network with 8 convolutional layers with 32, 32, 64, 64, 128, 128, 256 and 256 filters respectively and max pooling operators every two convolutional layers, followed by two fully connected layers of size 256.

For the two neural networks we used Dropout with probability $0.5$ between the fully connected layers, and optimized the network with the Adam algorithm with learning rate $0.001$.

The successive timestep for the prequential learning $t_1, t_2, ..., t_s$ are 8, 16, 32, 64, 128, 256, 512, 1024, 2048, 4096, 8192, 16384 and 32768.

For the prequential code results in Table 1, we selected the best model, which was the VGG-like network.

**Prequential Learning on CIFAR.**  On CIFAR, we used five different models:

1. The uniform probability over the labels.
2. A fully connected network or Multilayer Perceptron (MLP) with two hidden layers of dimension 512.
3. A shallow network, with one hidden layer and width 5000.
4. A convolutional network (tinyCNN) with four convolutional layers with 32 filters, and a maxpooling operator after every two convolutional layers. Then, two fully connected layers of dimension 256. We used Dropout with probability $0.5$ between the fully connected layers.
5. A VGG-like network with 13 convolutional layers from (Zagoruyko, 2015). We trained this architecture with two learning procedures. The first one (VGGa) without batch-normalization and data augmentation, and the second one (VGGb) with both of them, as introduced in (Zagoruyko, 2015). In both of them, we used dropout regularization with parameter 0.5.

We optimized the network with the Adam algorithm with learning rate $0.001$.

For prequential learning, the timesteps $t_1, t_2, ..., t_s$ were: 10, 20, 40, 80, 160, 320, 640, 1280, 2560, 5120, 10240, 20480, 40960. The training results can be seen in Figure 2.

For the prequential code, all the results are in Figure 2. For the results in Table 1, we selected the best model for the prequential code, which was VGGb.

# E  Switching between models against the *catch-up phenomenon*

## E.1  Switching between model classes

The solution introduced by (Van Erven et al., 2012) against the catch-up phenomenon described in Section 3.4, is to *switch* between models, to always encode a data block with the best model at that point. That way, the encoding adapts itself to the number of data samples seen. The switching pattern itself has to be encoded.

**Algorithm 2** Prequential encoding

---

**Input:** data $x_{1:n}, y_{1:n}$, timesteps $1 = t_0 < t_1 < ... < t_S = n$
Alice transmits the random seed $\omega$ (if using stochastic optimization).
Alice encodes $y_{1:t_1}$ with the uniform code. This costs $t_1 \log_2 K$ bits. Bob decodes $y_{1:t_1}$.
**for** $s = 1$ **to** $S - 1$ **do**
    Alice and Bob both compute $\hat{\theta}_s = \hat{\theta}(x_{1:t_s}, y_{1:t_s}, \omega)$.
    Alice encodes $y_{t_s+1:t_{s+1}}$ with model $p_{\hat{\theta}_s}$. This costs $-\log_2 p_{\hat{\theta}_s}(y_{t_s+1:t_{s+1}}|x_{t_s+1:t_{s+1}})$ bits
    Bob decodes $y_{t_s+1:t_{s+1}}$
**end for**

---

Table 2: **Compression bounds by switching between models.** Compression bounds given by different codes on two datasets, MNIST and CIFAR10. The *Codelength* is the number of bits necessary to send the labels to someone who already has the inputs. This codelength *includes* the description length of the model. The *compression ratio* for a given code is the ratio between its codelength and the codelength of the uniform code. The *test accuracy* of a model is the accuracy of its predictions on the test set. For variational and prequential codes, we selected the model and hyperparameters providing the best compression bound.

| CODE | MNIST | | | CIFAR10 | | |
|---|---|---|---|---|---|---|
| | CODELENGTH (kbits) | COMP. RATIO | TEST ACC | CODELENGTH (kbits) | COMP. RATIO | TEST ACC |
| UNIFORM | 199 | 1. | 10% | 166 | 1. | 10% |
| VARIATIONAL | 24.1 | 0.12 | 95.5% | 89.0 | 0.54 | 61,6% |
| PREQUENTIAL | **4.10** | **0.02** | **99.5%** | 45.3 | 0.27 | **93.3%** |
| SWITCH | **4.05** | **0.02** | **99.5%** | **34.6** | **0.21** | 93.3% |
| SELF-SWITCH | **4.05** | **0.02** | **99.5%** | **34.9** | **0.21** | 93.3% |

Assume that Alice and Bob have agreed on a set of prediction strategies $\mathcal{M} = \{p^k, k \in \mathcal{I}\}$. We define the set of switch sequences, $\mathbb{S} = \{((t_1, k_1), ..., (t_L, k_L)), 1 = t_1 < t_2 < ... < t_L, k \in \mathcal{I}\}$.

Let $s = ((t_1, k_1), ..., (t_L, k_L))$ be a switch sequence. The associated prediction strategy $p_s(y_{1:n}|x_{1:n})$ uses model $p^{k_i}$ on the time interval $[t_i; t_{i+1})$, namely

$$p_s(y_{1:i+1}|x_{1:i+1}, y_{1:i}) = p^{K_i}(y_{i+1}|x_{1:i+1}, y_{1:i}) \quad (E.1)$$

where $K_i$ is such that $K_i = k_l$ for $t_l \leq i < t_{l+1}$. Fix a prior distribution $\pi$ over switching sequences (see (Van Erven et al., 2012) for typical examples).

**Definition 4** (Switch code). Assume that Alice and Bob have agreed on a set of prediction strategies $\mathcal{M}$ and a prior $\pi$ over $\mathbb{S}$. The *switch code* first encodes a switch sequence $s$ strategy, then uses the prequential code with this strategy:

$$L_s^{\text{sw}}(y_{1:n}, x_{1:n}) = L_\pi(s) + L_{p_s}^{\text{preq}}(y_{1:n}, x_{1:n}) = -\log_2 \pi(s) - \sum_{i=1}^{n} \log_2 p^{K_i}(y_i|x_{1:i}, y_{1:i-1}) \quad (E.2)$$

where $K_i$ is the model used by switch sequence $s$ at time $i$.

We then choose the switching strategy $s^*$ wich minimizes $L_s^{\text{sw}}(y_{1:n}, x_{1:n})$. We tested switching between the uniform model, a small convolutional network (tinyCNN), and a VGG-like network with two training methods (VGGa, VGGb) (Appendix D). On MNIST, switching between models does not make much difference. On CIFAR10, switching by taking the best model on each interval $[t_k; t_{k+1})$ saves more than $11$ kbits, reaching a codelength of $34.6$ kbits, and a compression ratio of $0.21$. The cost $L_\pi(s)$ of encoding the switch $s$ is negligible (see Table 2).

### E.2 Self-Switch: Switching between variants of a model or hyperparameters

With standard switch, it may be cumbersome to work with different models in parallel. Instead, for models learned by gradient descent, we may use the same architecture but with different parameter values corresponding obtained at different gradient descent stopping times. This is a form of regularization via early stopping.

Figure 3: **Compression with the self-switch method:** Results of the self-switch code on CIFAR with 2 different models: the shallow network, and the VGG-like network trained with data augmentation and batch normalization (VGGb). Performance is reported during online training, as a function of the number of samples seen so far. Top: test accuracy on a pack of data $[t_k; t_{k+1})$ given data $[1; t_k)$, as a function of $t_k$. Second: codelength per sample (log loss) on a pack of data $[t_k; t_{k+1})$ given data $[1; t_k)$. Third: difference between the prequential cumulated codelength on data $[1; t_k]$, and the uniform encoding. Bottom: compression ratio of the prequential code on data $[1; t_k]$. The catch-up phenomenon is clearly visible for both models: even if models with and without the self-switch have similar performances after a training on the entire dataset, the standard model has lower performances than the uniform model (for the 1280 first labels for the VGGb network, and for the 10,000 first labels for the shallow network), and the code length for these first labels is large. The self-switch method solves this problem.

Let $(p_\theta)_{\theta \in \Theta}$ be a model class. Let $\hat{\theta}_j(x_{1:k}, y_{1:k})$ be the parameter obtained by some optimization procedure after $j$ epochs of training on data $[1; k]$. For instance, $j = 0$ would correspond to using an untrained model (usually close to the uniform model).

We call *self-switch code* the switch code obtained by switching among the family of models with different gradient descent stopping times $j$ (based on the same parametric family $(p_\theta)_{\theta \in \Theta}$). In practice, this means that at each step of the prequential encoding, after having seen data $[1; t_k)$, we train the model on those data and record, at each epoch $j$, the loss obtained on data $[t_k; t_{k+1})$. We then switch optimally between those. We incur the small additional cost of encoding the best number of epochs to be used (which was limited to 10) at each step.

The catch-up phenomenon and the beneficial effect of the self-switch code can be seen in Figure 3.

The self-switch code achieves similar compression bounds to the switch code, while storing only one network. On MNIST, there is no observable difference. On CIFAR, self-switch is only 300 bits (0.006 bit/label) worse than full 4-architecture switch.