[Reviews · NeurIPS 2018]

Reviewer 1



Summary : The paper considers the problem of minimum description length of any model - resting on the conjecture (as supported by Solomonoff’s general theory of inference and the Minimum Description Length) that a good model must compress the date along with its parameters well. Thus generalization is heuristically linked to compressive power of the model. With their experimental results, the authors surprise showing good overall compressive power of Deep Learning models using the prequential coding approach. Relating to the "Chaitin’s hypothesis [Cha07] that “comprehension is compression” any regularity in the data can be exploited both to compress it and to make predictions", the authors focus only on the compression problem . Detailed Comments: Clarity: The paper in well written and presents a good survey of the related work. Originality: There is no theory in the paper. The paper presents an empirical study of the possible confirmation of the Chaitin’s hypothesis [Cha07] that DL generalizes as it compresses well too, i.e., it has small MDL. For whatever it is worth, it is an interesting approach to understand an important problem in DL community, viz. the generalization issue despite the huge number of parameters in a DL model. Quality and Significance: I have the following concerns towards the acceptance of this paper: (a) No Sound Theory : This is the biggest drawback. Beyond the two experiments presented it's not rigorously clear, that describing a model with less description complexity has direct connection to generalization. You can always use a over-parametrized model, and still get very good generalization (see https://arxiv.org/abs/1710.05468), and that over-parametrized model may not have good MDL. Perhaps, what we need to quantify is some inter layer quantity layer (like Tishby's paper: https://arxiv.org/abs/1503.02406), so that the representation is learned to be well-suited for classification. So the connection with MDL is not clear. (b) Experiments not reproduced carefully : There is further concern about Section 3.4. Prequential coding applying to non-sequential data (e.g MNIST) is equivalent to training ConvNet without shuffling the data batches. The result with MNIST at 99.5% is not state of the art. State of the art is around 99.8% (https://github.com/hwalsuklee/how-far-can-we-go-with-MNIST). (c) Lack of enough experiments : For the claim supporting prequential coding causing good generalization, a better experiment will be using other dataset, which is not well studied as MNIST and CIFAR. Comparing Vanilla ConvNet model and Prequential Coding ConvNet model without any explicit regularization will be fair. Prequential code may not work very well, since it doesn't have shuffling dataset ability, which might make neural network overfit in some order-based rules. Overall, I like the idea of applying Information Theory to Deep Learning, but description length may not be the right way, or atleast has not been explained and exposited rigorously with this submission. A good revision may be required. ****** In the light of the author-feedback after the review process, I agree that quite a few of my concerns have been addressed and hence the overall score and the score of reproducibility has been updated.

Reviewer 2



This paper studies deep learning models under the perspective of minimum description length principle. In MDL principle, the quality of the model is quantified by the minimum length of the sequence to transmit the data. The author study the traditional compression bounds for neural networks, and claimed that the traditional bounds are not efficient. Then they introduced s coding scheme called "prequential coding" and show that prequential coding can significantly reduce the description length. Overall, I think the quality and clarity of this paper is above the standard of NIPS. So I vote to accept this paper, but I also would like to see the authors provide some feedbacks to the following questions: 1. A key question is that how can MDL theory help us improve deep learning is missing. Originally MDL is a quantity that measures a statistical model and can be used to help us choose a proper model. In this paper, prequential coding was proposed to compute MDL, but can it help us select models? 2. A second question is, since prequential coding gives much more efficient compression bounds, compared to state-of-the-art network compression algorithms, I wonder can prequential coding provides some intuition to improve network compression algorithms. ******************************************************************************************* I'm happy that the author provide some useful intuition and discussions in the feedback. I'll keep my score as 7.

Reviewer 3



The paper applies the two-part description approach from the minimum description length principle (or the Kolmogorov structure function) to understand how to find a good "compressed neural network model" for given data. This approach provides an apparently more principled way to compressing a large number of parameters in a deep neural network, which has lots of practical interests. Although the theoretical development is somewhat shallow (and sometimes misleading), the overall idea is quite sound, and in the end, the proposed prequential description approach provides an impressive performance. The overall flow is somewhat anecdotal and sometimes confusing (and certainly not the most straightforward path to the conclusion), but the storyline is still interesting. Although this point is not clearly stated (which the authors should address in the revised manuscript), the paper considers the problem of finding an efficient description of the output sequence of a given input sequence, when the pair is related through a given model (a deep neural network). One sufficient method of such a description is a two-part method -- first, describe the model and then compress the output sequence according to the model (conditioned on the input). This is at best a heuristic method, and it is not surprising that the method in itself performs poorly in practice. As a more average-case alternative, a variational approach, once again can be decomposed into two parts of descriptions, solves the efficient two-part description for random data. Although it provides a significantly better compression performance, the accuracy of the description is poor. As a third alternative, the paper develops a two-part description based on blockwise prequential model selection, which achieves both high compression efficiency and test accuracy. Although there is not much of a theoretical justification (who knows why two-part description is the right way -- or even an answer to the problem of describing the output given the input), the experimental result is quite remarkable and brings lots of discussions on the interpretations of what's going on. In this sense, we'd better err on accepting a half-baked idea, instead of waiting for a fully grown theoretical development, which might never occur. The literature citation is somewhat kinky throughout, often omitting key papers and promoting subsequent, secondary sources. For example, how come Rissanen's papers are omitted, yet Grunwald's monograph appears in the abstract? Similarly, Kolmogorov and Chaitin should receive the same spotlight as Solomonoff.